# HSImul3R: Reconstructing Simulation-Ready Human-Scene-Interaction from Sparse Views

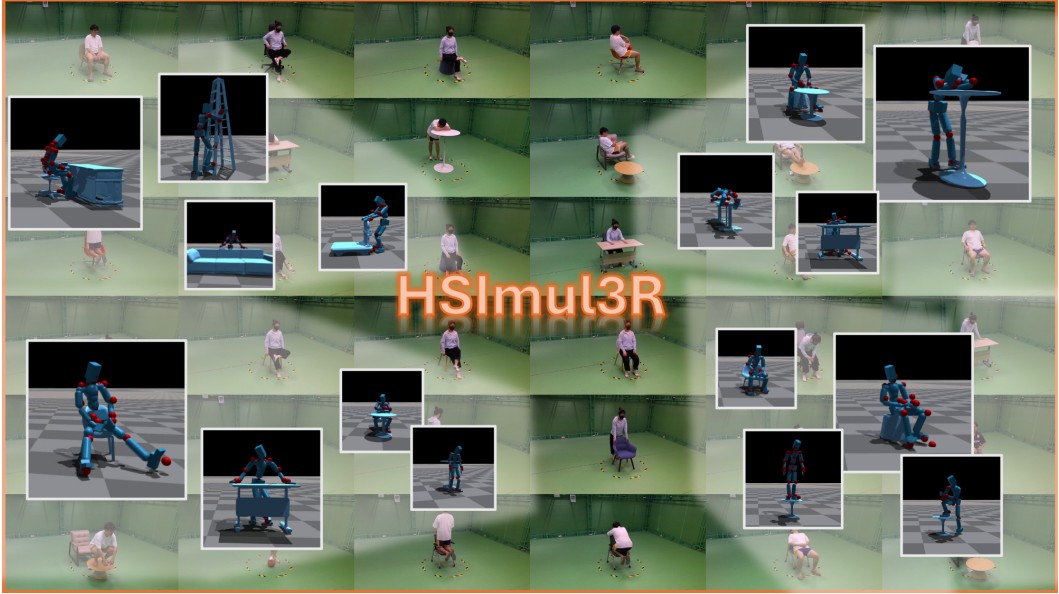

Figure 1: **Examples of results and data presented in HSImul3R.** Our approach enables simulation-ready 3D reconstruction of human–scene interactions from uncalibrated sparse-view inputs. In addition, we collect HSIBench, a dataset comprising 16-view synchronized captures of diverse human–scene interactions, covering a wide range of scene objects, human subjects, and motions.

## ABSTRACT

We present the first framework for simulation-ready 3D reconstruction of human–scene interactions (HSI) from sparse-view images. Prior approaches to 3D reconstruction are typically fragmented, focusing either on scene geometry or human motion, and rarely model their interactions. There are also recent attempts that reconstruct both jointly. However, they remain constrained by limited datasets or neglect the physical plausibility of interactions, and therefore fail to remain stable when deployed in simulators, which is a critical requirement for embodied AI. To address these challenges, we propose **HSImul3R**[1] with three key contributions. Specifically, firstly, we introduce **contact-aware interaction modeling** to enforce realistic human-scene coupling within the unified 3D world coordinate system by aligning generative 3D priors with reconstructed geometry. Secondly, we propose a **scene-targeted reinforcement learning** which learns to stabilize interactions in simulation through dual supervision on motion fidelity and object proximity. To further improve the stability of this HSI simulation, we design **direct simulation reward optimization (DSRO)**, a reward-driven fine-tuning scheme that improves scene reconstructions by assessing stability under both gravity and interactions. To support training and evaluation, we further collect **HSIBench**, a new dataset featuring diverse objects, human motions, and interaction scenarios. Extensive experiments demonstrate that HSImul3R achieves the first stable, simulation-ready HSI reconstructions and substantially outperforms existing methods.

---

[1]Pronunciation: / ˈsɪmjʊlə(r) /

# 1 INTRODUCTION

Embodied artificial intelligence has recently emerged as a prominent research direction, driven by the vision of integrating intelligent systems into human daily life through physically grounded agents. Unlike disembodied AI models that operate solely in virtual domains with topologies like clothing, embodied AI focuses on learning the transferable motions that enable agents to perceive, reason, and act in real-world environments. A central challenge in this field lies in modeling the interaction between humanoid robots and complex scenes, which requires understanding not only human motions, static spatial layouts, but also the stability of the interaction between human and scene components. In this context, reconstructing human-scene interactions (HSI) from images or videos has become an essential research direction, as it provides rich supervisory signals for learning how humans navigate, manipulate, and adapt to the surrounding scenes.

While significant advancements have been made in recent years, current reconstruction methods rarely model the coupling between humans and the environment. They can be broadly fragmented into three main directions: (1) *3D scene reconstruction* (Li et al., 2023b; Yan et al., 2024b; Ruan et al., 2025; Liu et al., 2025a; Huang et al., 2025b), where despite the progress in effecient 3D representations such as NeRF (Mildenhall et al., 2020) and 3D Gaussian splatting (Kerbl et al., 2023), most methods primarily emphasize modeling the surrounding scenes rather than human-centered interactions. Although extensions like HOSNeRF (Liu et al., 2023) incorporate human-centric interaction into reconstructed scenes, they restrict interactions to hand regions and often produce results that lack accuracy and realism. (2) *Human motion estimation* (Cai et al., 2023; Tian et al., 2023; Pavlakos et al., 2024; Li et al., 2025a), where methods have advanced toward robustness under occlusion and in human–object interactions (HOI) or human-human interactions (HHI) scenarios, yet they reconstruct solely the human motions and typically neglect the explicit coupling between humans and the surrounding environment. (3) *Interaction modeling* (Yang et al., 2022; Jiang et al., 2024a; Chen et al., 2024b; Fan et al., 2024; Pan et al., 2025), which relies heavily on SMPL-based (Loper et al., 2015) HSI or HOI datasets, *e.g.*, BEHAVE (Bhatnagar et al., 2022), CHAIRS (Jiang et al., 2023), HOMOTO (Lu et al., 2025). These datasets, however, are small in scale and diversity, resulting in poor generalization beyond their training domains, which constrains the generalizability of learned models to out-of-domain scenarios. Collectively, these fragmented directions demonstrate strong individual capabilities but fall short of explicitly reconstructing human–scene interactions.

More recently, advances in transformer-based architectures have enabled new possibilities and more effective frameworks. DUSt3R (Wang et al., 2024), for instance, introduces a pointmap representation with a joint optimization scheme that simultaneously predicts pointmaps, camera poses, and depth maps to reconstruct 3D scenes from uncalibrated multi-view images. Building upon this, HSfM (Müller et al., 2025) has sought to bridge the above gap by jointly reconstructing scene pointmaps, camera poses, and human motions in a unified global coordinate system. It incorporates human motion estimation into DUSt3R by proposing a unified pipeline that reconstructs scene point clouds, camera poses, and human motions under a consistent global coordinate system. While this marks progress toward combining human and scene information, HSfM still does not explicitly enforce interaction constraints and is not simulation-ready for embodied AI applications: **(1)** The quality of DUSt3R-based reconstructions remains limited, and without explicit physical constraints, HSfM often produces unstable 3D shapes lacking realistic geometry and structure. Such outputs struggle to remain stable and balanced under gravity, as in the simulator. **(2)** Because the optimization process is driven solely by 2D projections, HSfM lacks mechanisms to enforce physically plausible human–scene interactions. As a result, penetration artifacts and unrealistic collisions frequently arise when transferring reconstructions into a simulator.

In this paper, we introduce a new **H**uman-**S**cene-**I**nteraction si**mul**ation-ready **3D R**econstruction from sparse-view images, dubbed **HSImul3R**. Our method is built upon HSfM and comprises three novel components to address the aforementioned challenges.

*(1) Contact-Aware Interaction Modeling*: To overcome the topological inconsistencies of HSfM, we explicitly inject 3D structural priors derived from image-to-3D generative models. Specifically, we leverage the pre-trained MIDI model (Huang et al., 2025a) to reconstruct plausible 3D scenes from input images, and integrate these reconstructions with HSfM outputs to refine structural accuracy. Beyond scene geometry, we further introduce a contact-aware interaction module that enforces physically consistent human–scene coupling by pulling humans closer to surfaces where contact is expected, and pushing them apart when penetrations occur.

*(2) Scene-Targeted Reinforcement Learning*: To improve stability once the reconstructed HSI is placed into a physics-based simulator (Makoviychuk et al., 2021), we design a reinforcement learning policy that blends two supervisory signals: one for human keypoint tracking and another for minimizing distances between humans and relevant objects. This dual-objective formulation enables the model to dynamically refine interaction strategies in simulation, while ensuring that the resulting states remain faithful to the original human motion.

*(3) Direct Simulation Reward Optimization (DSRO)*: While reinforcement learning stabilizes human–scene interactions, the underlying MIDI-based scene reconstructions are not always topologically accurate, which can still lead to simulation failures. Inspired by DSO (Li et al., 2025b), we reformulate this challenge as a reward-driven optimization problem and propose DSRO to fine-tune the pre-trained MIDI model. Unlike DSO, which evaluates stability solely under gravity, our approach leverages the proposed contact-aware interaction modeling and scene-targeted reinforcement learning to assess stability with respect to both gravity and human–scene interactions. This allows the system to better capture realistic dynamics and avoid failures caused by implausible contacts or penetrations. To support this training, we further collect **HSIBench**, a new dataset comprising 19 objects, more than 50 motion sequences, and recordings from two male and one female participants, totaling 300 unique subjects.

We conduct extensive experiments to evaluate HSImul3R against state-of-the-art baselines in terms of simulation stability, post-simulation human motion quality, and improvements in image-to-3D generation through DSRO fine-tuning. Experimental results demonstrate that HSImul3R is the first approach to achieve stable, simulation-ready reconstructions of human–scene interactions, offering robust performance across diverse scenarios and significantly outperforming existing techniques. An overview of our method is provided in Fig. 1.

## 2 RELATED WORKS

**3D Scene Reconstruction.** Early approaches are dominated by geometry-based methods, such as structure-from-motion (Schönberger & Frahm, 2016) and multi-view stereo (Seitz et al., 2006), which estimate camera poses and dense geometry from multiple views. With the rise of deep learning, data-driven approaches emerge, including monocular depth prediction (Yang et al., 2024a;b) and learning-based multi-view stereo (Huang et al., 2018), enabling reconstruction from sparse or unstructured imagery. Other works adopt explicit 3D representations such as voxels (Song et al., 2017; Liu et al., 2025b), point clouds (Dai et al., 2017; Xie et al., 2020), and meshes (Nie et al., 2020), often optimized through differentiable rendering. More recently, implicit neural representations, such as signed distance functions (Park et al., 2019), occupancy fields (Bian et al., 2025), neural radiance fields (Li et al., 2023b; Xie et al., 2024a), and explicit but differentiable formulations like 3D Gaussian Splatting (Kerbl et al., 2023; Xie et al., 2025a), become central to high-quality scene modeling. Beyond static reconstruction, dynamic scene modeling (Yan et al., 2024b; Xie et al., 2025b) expands these methods to time-varying environments. In parallel, recent works such as Dust3R (Wang et al., 2024) and VGGT (Wang et al., 2025) introduce pre-trained transformers that enable end-to-end 3D reconstruction directly from uncalibrated and unlocalized images, eliminating the need for expensive post-optimization.

**Physically-sounded Modeling.** Recent works have sought to embed physical soundness into modeling, which can be broadly categorized into three paradigms. Physics-constrained and physics-integrated generation methods unify simulation and content creation by leveraging simulation-derived losses or physical priors. For example, PhyRecon (Ni et al., 2024) ensures stable scene reconstruction, Atlas3D (Chen et al., 2024a) and BrickGPT (Pun et al., 2025) produce self-supporting structures, and DSO (Li et al., 2025b) or PhysDeepSDF (Mezghanni et al., 2022) align generators with simulation feedback. PhysGaussian (Xie et al., 2024b) evolves Gaussian splats via continuum mechanics, while PhyCAGE (Yan et al., 2024a), VR-GS (Jiang et al., 2024b), and GASP (Borycki et al., 2024) optimize assets through MPM; PAC/iPAC-NeRF (Li et al., 2023a; Kaneko, 2024) jointly learn geometry and physical parameters to bridge reconstruction and simulation. This approach also extends to interactive contexts: PhyScene (Yang et al., 2024c) generates simulation-ready environments, PhysPart (Luo et al., 2024a) models functional parts for robotics and fabrication, and DreMa (Barcellona et al., 2025) produces manipulable, physics-grounded world models.

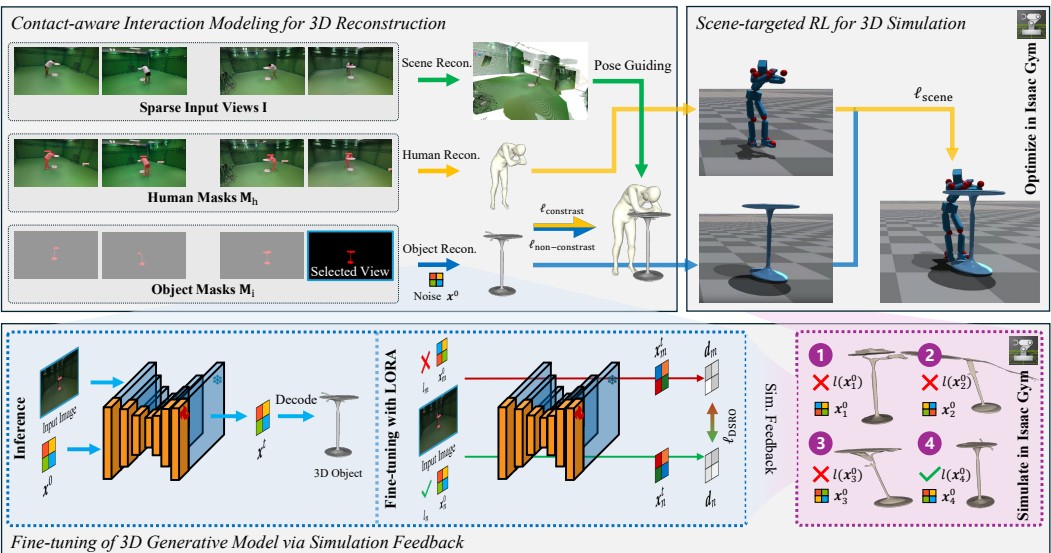

Figure 2: **Overview of HSImul3R.** Given sparse-view inputs $\mathbf{I}$, we achieve simulation-ready reconstruction of human–scene interactions through three key components: **(1)** a contact-aware interaction model that leverages 3D generative priors to optimize interactions in world coordinates, **(2)** a scene-targeted reinforcement learning strategy that enhances stability when integrating reconstructions into physics-based simulators, and **(3)** a direct simulation reward optimization that fine-tunes the pre-trained 3D generative model using feedback from failed simulations (Type 1: objects not stabilizing under gravity; Type 2: objects failing to stabilize during human interaction; Type 3: objects stabilizing but without meaningful interaction; Type 4: objects with stable interaction).

**Human Simulation Imitating.** Recent advances in physics-based humanoid simulation fall into three directions. Robust motion imitation builds on RL frameworks such as DeepMimic (Peng et al., 2018) and AMP (Peng et al., 2021), extended by PHC (Luo et al., 2023) for long-horizon resilience and DiffMimic (Ren et al., 2023a) with differentiable physics. More recent methods leverage human demonstrations for adaptive whole-body imitation, including locomotion and manipulation, as in HumanPlus (Fu et al., 2024) and TWIST (Ze et al., 2025). Generalizable control is advanced by PULSE (Luo et al., 2024c), which provides compact latent spaces for versatile skills, HOVER (He et al., 2024), which unifies multiple control modes, and diffusion-based frameworks such as CLoSD (Tevet et al., 2025) and InsActor (Ren et al., 2023b), which integrate generative planning with physics-based execution for multi-task behaviors. Interactive skills cover dynamic human-object interactions and complex benchmarks: PhysHOI (Wang et al., 2023) and Omni-Grasp (Luo et al., 2024b) enable dexterous manipulation, SMPLOlympics (Luo et al., 2024d) and HumanoidOlympics (Luo et al., 2024e) provide sports environments, Half-Physics (Siyao et al., 2025) bridges kinematic avatars with physics, ImDy (Liu et al., 2025c) exploits imitation-driven simulation, and ASAP (He et al., 2025) improves fidelity by aligning dynamics with demonstration trajectories.

## 3 METHODOLOGY

Given $J$ sparse view images, HSImul3R is designed to reconstruct 3D simulation-ready human-scene-interactions. We set $J = 4$ for experiments reported in this paper. As illustrated in Fig. 2, the reconstruction procedure comprises two parts: 3D scene reconstruction via DUSt3R (Wang et al., 2024) and multi-view human motion estimation via 4D-Humans (Goel et al., 2023)followed by physics-based human–scene interaction simulation in Isaac Gym (Makoviychuk et al., 2021). In the subsequent sections, we first present the preliminaries underlying our method in Sec. 3.1, and then detail the three key components of HSImul3R: (1) contact-aware interaction modeling during reconstruction with 3D generative priors (Sec. 3.2), (2) scene-targeted reinforcement learning for stable interaction simulation (Sec. 3.3), and (3) direct simulation reward optimization to fine-tune the generative model using simulation feedback (Sec. 3.4).

## 3.1 PRELIMINARIES

**DUSt3R (Wang et al., 2024).** Recently, DUSt3R introduced a framework for 3D reconstruction that regresses point maps and employs a global alignment strategy to jointly predict depth maps and camera poses. Specifically, given a set of input images $\mathbf{I} = I_0, I_1, ..., I_J$, DUSt3R applies a ViT-based network that takes a pair of image frames $I_n, I_m$ $(n, m \in [0, J])$ to estimate the corresponding point maps $P_n^e, P_m^e \in \mathbb{R}^{H \times W \times 3}$ with respect to the coordinate system of frame $n$, along with confidence maps $C_n^e, C_m^e \in \mathbb{R}^{H \times W}$. Here, $e = (m, n)$ denotes the selected image pair. Aggregating point maps and confidence maps across selected pairs, DUSt3R builds a connectivity graph $\mathcal{G}(\mathcal{V}, \mathcal{E})$, where $\mathcal{V}$ corresponds to the $N$ images and $\mathcal{E}$ to the chosen image pairs $e$.

After collecting all pairwise point maps, DUSt3R performs a global alignment optimization to recover the depth maps $\mathbf{D} = D_0, D_1, ..., D_J$ and camera poses $\pi_0, \pi_1, ..., \pi_J$:

$$\arg\min_{D, \pi, \sigma} \sum_{e \in \mathcal{E}} \sum_{n \in e} C_n^e ||D_n - \sigma_e \cdot F_e(\pi_n, P_n^e)||_2^2, \quad (1)$$

where $\sigma = \sigma_e, e \in \mathcal{E}$ denotes the edge-wise scale factors, and $F_e(\pi_n, P_n^e)$ projects the predicted point map $P_n^e$ to view $n$ under camera pose $\pi_n$ to produce the corresponding depth. This objective enforces geometric alignment across frame pairs, ensuring cross-view consistency in the estimated depth maps after optimization.

**Human Structure from Motion (HSfM) (Müller et al., 2025).** Although DUSt3R achieves high-quality 3D reconstruction from uncalibrated images, it remains limited in handling human-centric elements such as pose recovery. To overcome this, HSfM (Müller et al., 2025) integrates DUSt3R with the traditional Structure-from-Motion (SfM) framework to jointly reconstruct scenes and human meshes. Specifically, given an image set $\mathbf{I} = I_0, I_1, ..., I_J$, HSfM first applies the pre-trained DUSt3R model, large-language segmentation model (Ravi et al., 2024), a multi-view human motion estimator (Goel et al., 2023), and a human keypoint detector (Xu et al., 2022) to obtain, respectively, 3D scene reconstructions $\mathcal{R}_{\text{scene}}$ (*i.e.*, rotation $R$, translation $t$, intrinsics $K$, depth map $D$, and pointmap $P$), human masks $M_h$, human poses (*i.e.*, orientation $\phi$, translation $\gamma$, pose parameters $\theta$, and shape $\beta$), and 2D human keypoints $J_{2D}$. These outputs are then jointly optimized through (1) bundle adjustment via human keypoint $J_{2D}$ and (2) global scene optimization, aligning humans, scenes, cameras, and scales into a unified coordinate system.

## 3.2 CONTACT-AWARE INTERACTION MODELING FOR 3D RECONSTRUCTION

Our approach for 3D HSI reconstruction is built upon HSfM (Müller et al., 2025). However, despite its advances, HSfM suffers from two critical issues: (1) it inherits DUSt3R's tendency to generate unrealistic scene structures (see Fig. 10 in Appx. B), and (2) it constrains human–scene interactions to 2D projections, both of which lead to instability in simulation. This limitation is expected, as HSfM imposes no explicit 3D constraints and sparse-view reconstruction is inherently ambiguous, particularly when human subjects occlude parts of the scene. To overcome these challenges, we introduce contact-aware interaction modeling that leverages pre-trained 3D generative priors.

**Image-to-3D Generative Model.** As shown in Fig. 2, we first enhance the reconstruction by incorporating a pre-trained image-to-3D generative model (Huang et al., 2025a) as an additional geometric prior to refine the outputs of HSfM. Concretely, for each object present in the scene, we automatically identify the input image $I_n, n \in [0, J]$ where the object is most prominently visible. Using SAM (Kirillov et al., 2023), we extract its segmentation mask $M_i$, and then employ the pre-trained MIDI model to generate a high-fidelity 3D representation. This synthesized object geometry replaces the corresponding reconstructed points in the original scene, thereby correcting structural inaccuracies and yielding a more realistic reconstruction:

$$\mathcal{R}_{\text{scene}} := \{\texttt{MIDI}(I_n[M_i]), i \in [0, O]\}, \quad (2)$$

where $\mathcal{R}_{\text{scene}}$ denotes the refined 3D scene and $O$ is the total number of objects.

**Contact-aware Interaction Modeling.** While the image-to-3D generative model provides refined scene reconstructions with more realistic topology, it does not account for interactions with the recovered human mesh. This limitation is expected, as the model lacks explicit 3D constraints to prevent interpenetration between humans and objects. Moreover, aligning human and scene geometry

through 2D projections alone is inherently unreliable, given challenges such as occlusions, sparse viewpoints, and incomplete 3D information. Importantly, penetration artifacts become particularly problematic in simulation: even minor inconsistencies in 3D space can manifest as severe collisions between body parts and objects, ultimately leading to unstable or failed simulations. To address this issue, we optimize the position of the recovered human and generated objects . Specifically, if the object and human are not in contact, we optimize their positions via:

$$\ell_{\text{non-contact}} = \frac{1}{|H_p|} \cdot \sum_{1 \leq j \leq N_o} ||\mu_i^h - \mu_j^o||_2 + \frac{1}{N_o} \cdot \sum_{j=1}^{N_o} \min_{i \in H_p} ||\mu_j^o - \mu_i^h||_2, \tag{3}$$

where $H_P$ denotes the human body part closest to the object, and $N_o$ is the number of vertices on the object, and $\mu_j^o$ and $\mu_i^h$ represent the 3D positions of object and human vertices, respectively. When the object is in contact with the human, we instead apply:

$$\ell_{\text{contact}} = \frac{1}{|H_p|} \cdot \sum_{i \in H_p} \max(0, -\delta(\mu_i^h)), \tag{4}$$

where $\delta(\cdot)$ denotes the signed distance function, measuring the penetration depth of the human vertex $\mu_i^h$ relative to the object surface.

### 3.3 SCENE-TARGETED REINFORCEMENT LEARNING FOR 3D SIMULATION

After reconstructing human–scene interactions in 3D space, the next step is to simulate them and ensure stable dynamics between humans and objects. To this end, we leverage the pre-trained PHC (Luo et al., 2023) model to retarget reconstructed human poses onto a humanoid robot within Isaac Gym (Makoviychuk et al., 2021). However, directly simulating the raw reconstructions often fails to yield stable interactions (see Fig. 4). In many cases, the humanoid inadvertently displaces nearby objects, leaving them separated from the body and resting independently on the ground. This instability arises because conventional 3D reconstructions do not account for interaction forces or verify whether human poses and object placements are physically realizable in a stable configuration. To address this, we extend PHC by adding a scene-targeted supervision signal alongside its original human keypoint tracking. While PHC was designed for dynamic motions, we adapt it to static poses by replicating each pose across the temporal dimension to fit the network.

Specifically, we introduce an additional objective that enforces spatial proximity between the humanoid and the relevant scene objects, thereby encouraging physically plausible and stable contact during simulation. The loss averages the Euclidean distance between human contact keypoints $k_j^h$ and their corresponding nearest object surface points $\mu_i^o$.

$$\ell_{\text{scene}} = \frac{1}{N_{\text{contact}} \cdot N_{\text{surf}}} \cdot \sum_{j=1}^{N_{\text{contact}}} \sum_{i=1}^{N_{\text{surf}}} ||\mu_i^o - k_j^h||_2^2, \tag{5}$$

where $N_{\text{contact}}$ is the number of contacts between the human and scene objects, and $N_{\text{surf}}$ denotes the number of sampled object surface points within the local contact region.

### 3.4 FINE-TUNING OF 3D GENERATIVE MODEL VIA SIMULATION FEEDBACK

Nonetheless, even with our scene-targeted reinforcement learning, the percentage of stable simulations remains unsatisfactory (see Tab. 1). As presented in Fig. 3, we observe that this problem largely stems from the inconsistent quality of the MIDI-based 3D generative prior, for two main reasons: (1) generated objects often contain structural defects, especially in slender geometries. For example, tables or chairs may be missing legs, making them unstable in the simulator even without interaction; and (2) severe occlusion by the human in the input images, which frequently happens, often results in generated objects exhibiting artifacts, such as surface distortions or unwanted bumps. Together, these limitations make it difficult for the humanoid to establish stable and physically plausible contact during simulation.

A straightforward strategy to alleviate this problem would be to finetune the pre-trained MIDI model. However, doing so raises the challenge of defining suitable supervision signals to directly address

Table 1: **Quantitative comparison regarding simulation stability of human-scene-interaction and the quality of human motions.**

| Method | Stability-HSI (%) ↑ | | | Scene Penetration - 3D (%) ↓ | Human Motion Quality | |
|--------|------|--------|------|------------------------------|-------------|-------------|
| | Easy | Medium | Hard | | W-MPJPE ↓ | PA-MPJPE ↓ |
| HSfM | 10.52 | 4.50 | 2.66 | 69.51 | 5.02 | 2.79 |
| **V1** | 13.96 | 8.81 | 4.17 | 77.12 | 6.18 | 3.20 |
| **V2** | 39.56 | 22.71 | 7.05 | - | 4.91 | 2.71 |
| **V3** | 42.57 | 23.84 | 10.18 | - | 4.60 | 2.42 |
| **V4** | 29.56 | 16.62 | 5.17 | - | 4.57 | 2.39 |
| Ours | **53.68** | **30.56** | **13.92** | **22.9** | **4.09** | **2.17** |

these structural and occlusion-related issues. Recent advances such as Diffusion-DPO (Wallace et al., 2024) offer one potential direction by leveraging pairwise human preference feedback as optimization signals, thereby aligning generative models with human judgments. Yet, the requirement for large-scale preference data makes this approach difficult to scale in practice. To overcome this bottleneck, DSO (Li et al., 2025b) proposes a direct reward optimization framework that eliminates reliance on 3D ground-truth data while still providing effective supervision for generative refinement.

**Direct Simulation Reward Optimization.** Building on these insights, we introduce Direct Simulation Reward Optimization (DSRO), a novel approach that leverages physics-based simulation feedback as a supervision signal for refining 3D object generation. Unlike preference-based methods that rely on human annotations or 3D ground truth, DSRO directly exploits the outcome of the simulation to assess the physical plausibility of generated objects and their interactions with humans.

Formally, we follow DSO and define the DSRO objective as:

$$\ell_{\text{DSRO}} = -T\mathbb{E}_{I\sim\mathcal{I},x_0\sim\mathcal{X}_I,t\sim\mu(0,T),x_t\sim q(x_t|x_0)}[w(t)\cdot(1-2\cdot l(x_0))||\epsilon - \epsilon_\theta(x_t,t)||^2_2]], \quad (6)$$

where $I$ denotes an image sampled from the training dataset $\mathcal{I}$, $\mathcal{X}$ corresponds to its MIDI-generated 3D representation, and $l(\cdot)$ encodes the stability feedback obtained from simulation. Crucially, in contrast to DSO (Li et al., 2025b), which measures stability solely based on whether an object remains upright under gravity, our formulation defines $o_{\text{stable}}$ as follows:

$$l(x_0) = \begin{cases} 1, & \text{if stable} \\ 0, & \text{otherwise,} \end{cases} \quad (7)$$

where stability is determined according to three criteria: (1) the object must remain upright and physically stable under gravity within the simulator, (2) it must achieve a stable final state for the reconstructed scene, and (3) the interaction must involve actual contact rather than the object resting independently on the ground.

**HSIBench.** To enable effective fine-tuning of the pre-trained MIDI generative model, we construct a dedicated benchmark dataset, HSIBench, tailored for human–scene interaction (HSI). The dataset is built by systematically capturing interaction scenarios involving two volunteers (one male and one female) engaging with a diverse set of objects, including eight chairs, three tables, and three sofas. In total, we record 300 distinct HSI cases, with each case captured from 16 different viewpoints to provide rich multi-view supervision. Representative examples are illustrated in Fig. 6 to Fig. 9 in Appx. A). For every captured case, we run our full reconstruction and simulation pipeline, as described in Sec. 3.2 and Sec. 3.3, 12 times under different random seeds. This procedure ensures variability in the simulation outcomes and allows us to systematically collect the training signals needed for fine-tuning.

## 4 EXPERIMENTS

We evaluate HSImul3R across three dimensions: reconstruction fidelity, simulation stability, and the impact of fine-tuning with the proposed DSRO. We also benchmark against existing methods and perform ablation studies to assess the contribution of each component.

**Implementation Details.** Our approach is developed on top of HSfM (Müller et al., 2025) and PHC (Luo et al., 2023). For training, we adopt AdamW (Loshchilov & Hutter, 2017) as the optimizer and fine-tune the pre-trained MIDI model using LoRA (Hu et al., 2022). Specifically, we set the

Table 2: **Quantitative comparison regarding image-to-3D generation quality.**

| Method | Stability-HSI (%) ↑ | | | Stability-Gravity (%) ↑ | Chamfer Distance ↓ | F-Score ↑ |
|---|---|---|---|---|---|---|
| | Easy | Medium | Hard | | | |
| MIDI | 29.56 | 16.62 | 5.17 | 79.19 | 0.198 | 81.95 |
| DSO* | 38.75 | 25.91 | 7.88 | 87.23 | 0.191 | 86.26 |
| Ours | **53.68** | **30.56** | **13.92** | **91.50** | **0.173** | **88.25** |

LoRA rank to 64, use a batch size of 32, and a learning rate of 0.00001. The model is trained for a total of 6000 steps on four NVIDIA A100 GPUs.

**Baseline Methods.** Since our method presents the first approach for simulation-ready reconstruction of human–scene interactions from uncalibrated sparse-view inputs, we primarily compare its performance against HSfM (Müller et al., 2025), which is the first and only technique to reconstruct 3D HSI under sparse-view settings. Additionally, considering that there is no other dedicated method existing for this task, we further compare with various alternatives: **(V1)** a simple baseline that integrates HSfM with MIDI (Huang et al., 2025a) and feeds the resulting reconstruction into the simulator; **(V2)** using the reconstruction from Sec. 3.2 directly in the simulator without applying the scene-targeted distance minimization of Eq. 5; **(V3)** Replacing our object-surface distance computation with a center-point distance following CLoSD (Tevet et al., 2025); **(V4)** Obtain the simulated reconstruction directly via Sec. 3.2 and Sec. 3.3 without fine-tuning the generative model using simulation feedback via the proposed DSRO.

We also compare with the MIDI (Huang et al., 2025a) and DSO (Li et al., 2025b) in terms of the geometric quality of the generated scene objects, as well as stability under both gravity-only and HSI scenarios. For fairness, we fine-tune DSO on the pre-trained MIDI model rather than its originally used Trellis (Xiang et al., 2025) model.

**Evaluation Metrics.** We first evaluate the penetration ratio in the reconstructed 3D HSI scenes. Next, we assess the stability of simulated human–scene interactions using the metric Stability-HSI, which considers three factors: (1) object stability under gravity, (2) whether the HSI scene reaches a stable state in the simulator, and (3) whether the final state preserves meaningful human–scene interactions. Finally, we evaluate the quality of simulated human motion by extracting it from the final state and comparing it to the ground truth. Following HSfM, we report W-MPJPE for accuracy in the world coordinate system and PA-MPJPE for local pose precision.

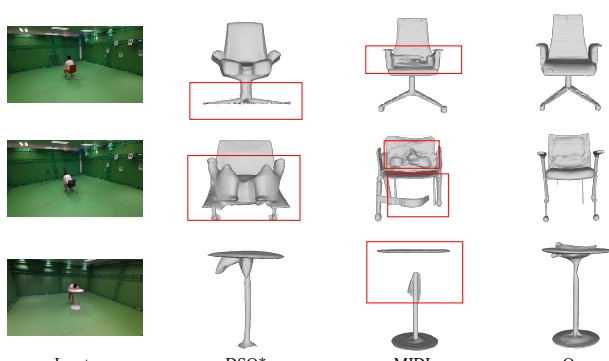

Figure 3: **Qualitative comparison regarding image-to-3D object reconstruction.**

For reconstructed 3D scene objects, we measure geometric quality using Chamfer Distance and F-Score, while physical plausibility is evaluated through "Stability-HSI" and "Stability-Gravity".

**Evaluation Datasets.** We perform both quantitative and qualitative evaluations on our collected HSIBench dataset. To assess HSI simulation stability across different scenarios, we divide HSIBench into three levels of difficulty, *i.e.*, easy, medium, and hard, based on interaction complexity.

## 4.1 RESULTS AND ANALYSIS

**Quantitative Evaluations.** We first present quantitative comparisons of HSI reconstruction and simulation quality in Tab. 1. As shown, our method significantly outperforms the only existing baseline, HSfM, as well as the ablated variants, across all evaluated metrics. This demonstrates both the overall effectiveness of our approach and the contribution of the proposed components. Note that V1, V2, and V3 do not report scene penetration percentages, as their 3D reconstruction is identical to that of V1. We then report quantitative results on image-to-3D generation quality in Tab. 2. Importantly, our method (incorporating DSRO) achieves improved physical plausibility and interaction stability, along with superior geometric accuracy.

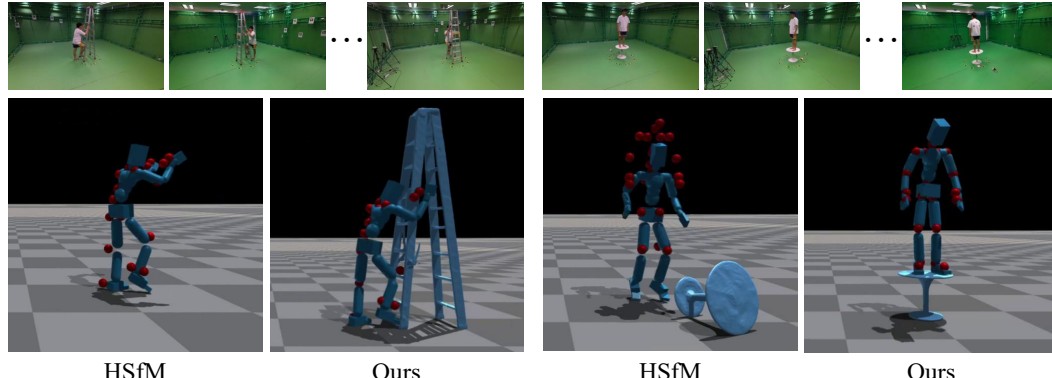

Figure 4: **Qualitative comparisons with HSfM .** Due to challenges such as (1) penetration issues and (2) inaccurate scene-object structures with geometric distortions, HSfM often struggles to achieve stable interactions in the simulator, frequently leading to unintended object displacement.

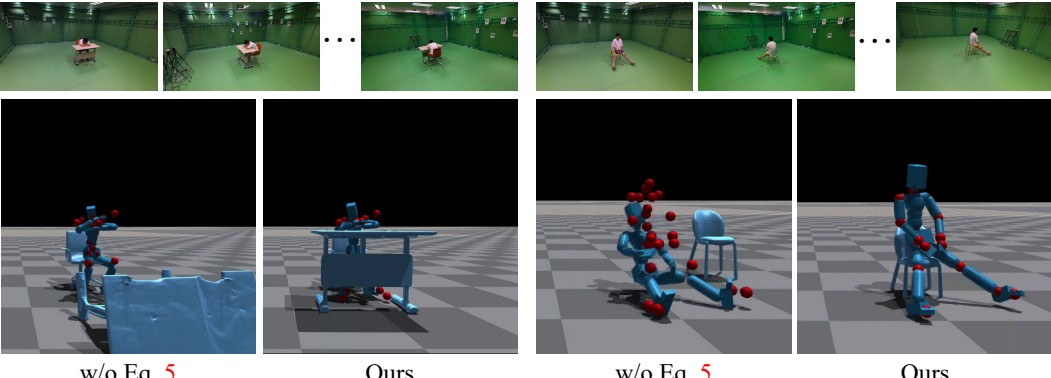

Figure 5: **Ablation studies on Eq. 5.** Without the proposed scene-targeted RL, the simulation often results in unintended object displacement and fails to maintain stable interactions.

**Qualitative Evaluations (1)** In Fig. 4, we present the qualitative comparisons with HSfM. Specifically, we apply Poisson reconstruction to the point maps generated by HSfM and place the reconstructed objects into the simulator for evaluation. As shown, HSfM often fails to produce stable human–object interactions: the human frequently kicks objects away and ends up standing alone. In contrast, our method consistently achieves stable interaction states within the simulation. **(2)** Fig. 3 further compares our approach with DSO and MIDI in terms of image-to-3D reconstruction quality. Both baselines struggle to recover accurate structures and often introduce geometric distortions, which in turn lead to instability during simulation. By contrast, our DSRO fine-tuned model mitigates these issues, yielding more structurally faithful and stable reconstructions.

**Analysis of Scene-targeted 3D Simulation.** Fig. 5 presents the ablation study on the scene-targeted 3D simulation loss defined in Eq. 5. Results indicate that removing the distance-minimization term destabilizes the humanoid, leading to exaggerated motions and often kicking objects away.

## 5 CONCLUSIONS

In this work, we introduced HSImul3R, the first framework for reconstructing simulation-ready human–scene interactions from uncalibrated sparse views. Our approach incorporates a contact-aware interaction model to mitigate human–scene penetration issues in 3D reconstruction, a scene-targeted reinforcement learning strategy to promote stable interactions within the simulator, and a direct simulation reward optimization scheme that leverages simulation feedback to fine-tune the image-to-3D generative model, thereby improving simulation success rates. To support both training and evaluation, we also collected the HSIBench dataset. Extensive experiments demonstrate that HSImul3R achieves high-fidelity results, delivering both stable simulations and high-quality image-to-3D reconstructions, and significantly outperforms existing state-of-the-art methods.

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

# APPENDIX

## ETHICS STATEMENT

This work uses only publicly available datasets and data collected under controlled conditions, without involving personal or sensitive information. Our framework is intended for advancing embodied AI and simulation research. We caution against potential misuse for surveillance or deceptive content generation, and release all resources strictly for academic purposes.

## REPRODUCIBILITY STATEMENT

We place strong emphasis on reproducibility, providing detailed descriptions to facilitate replication and validation. Details of the methodology and experimental setup are presented in Sec. 3 and Sec. 4, respectively. Upon acceptance, we commit to making the code, pretrained model weights, and comprehensive documentation publicly available to facilitate reproducibility and future research.

## USAGE OF LARGE LANGUAGE MODELS

During the preparation of this manuscript, large language models were employed solely as writing aids. They assisted in checking grammar, improving sentence structure, and suggesting stylistic alternatives. All methodological details, experimental results, and conclusions were developed exclusively by the authors. The outputs generated by the models were critically reviewed, and only author-verified edits were incorporated into the final version.

## A    VISUALIZATIONS OF HSIVBENCH

Figs. 6–9 illustrate example visualizations from our collected HSIVBench dataset. HSIVBench consists of 300 human–scene interaction video pairs, each captured from 16 synchronized views. In total, the dataset includes 19 distinct objects (e.g., eight chairs, four tables, five sofas, one pushcart, and one staircase) that are commonly encountered in daily life, three participants (two male and one female), and more than 50 human motions.

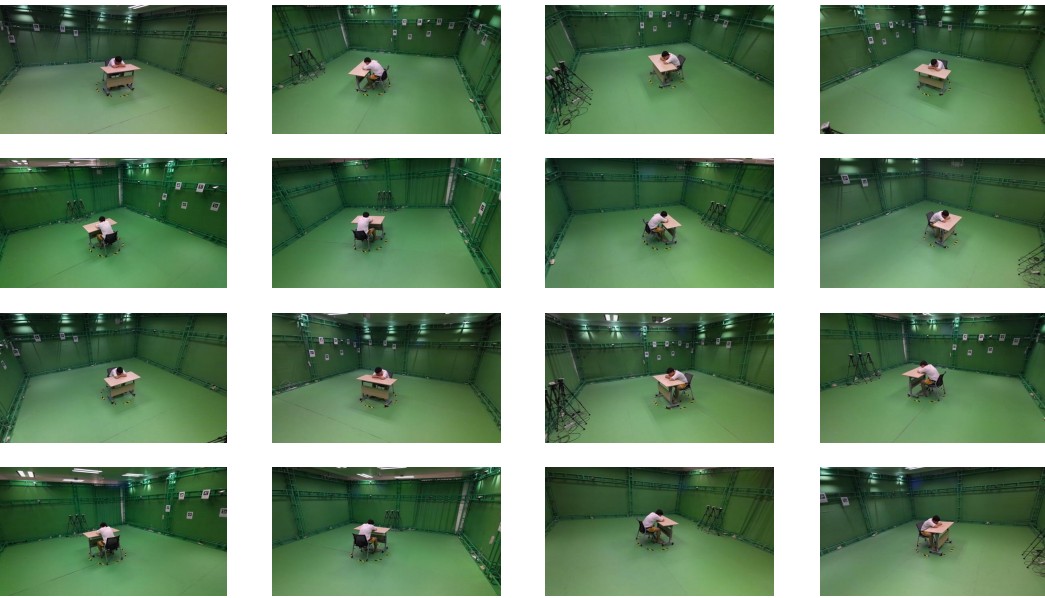

Figure 6: **Examples of HSIVBence from 16 views.**

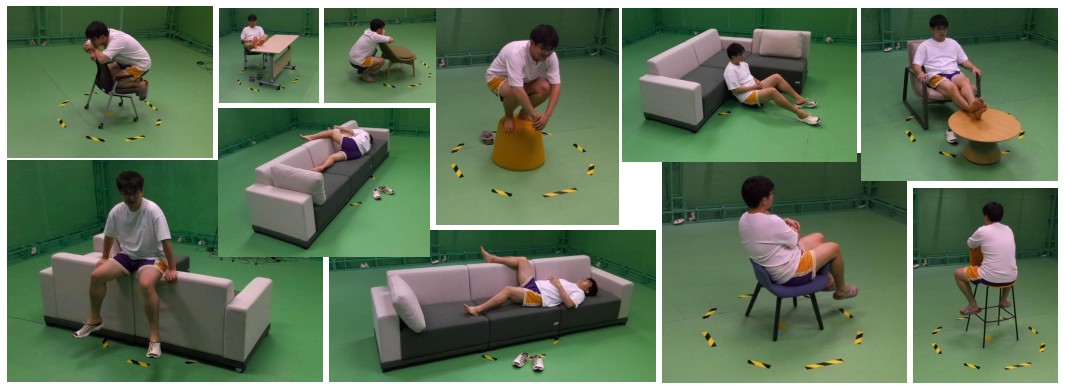

Figure 7: **Examples of different human-scene-interactions captured in HSIVBench.**

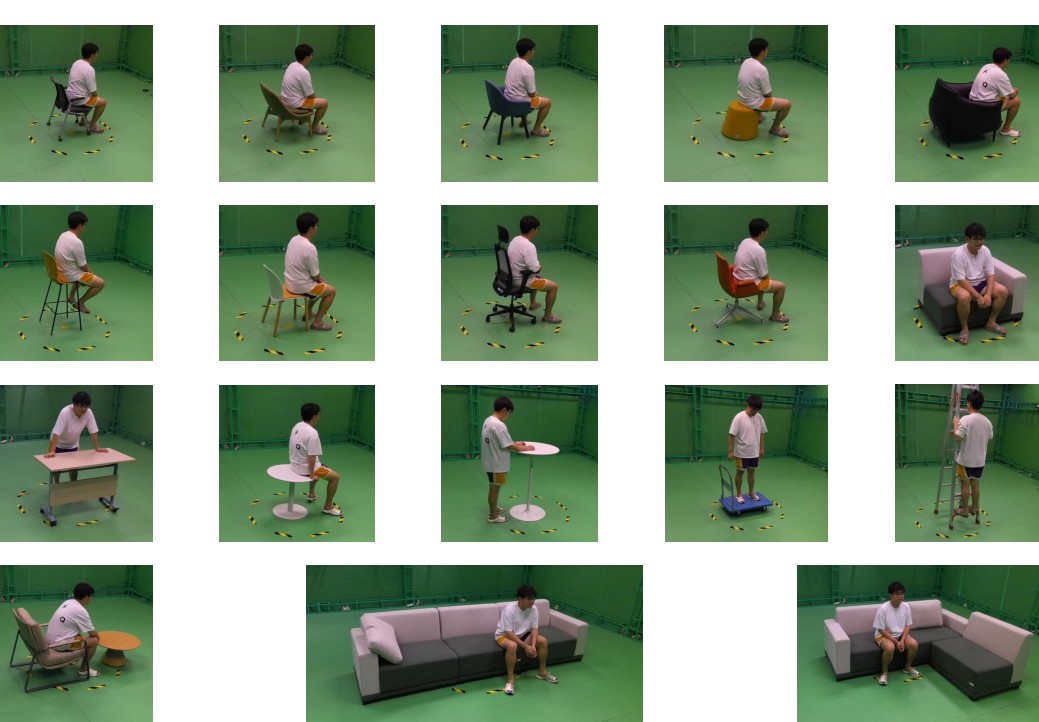

Figure 8: **Examples of different objects captured in HSIVBench.**

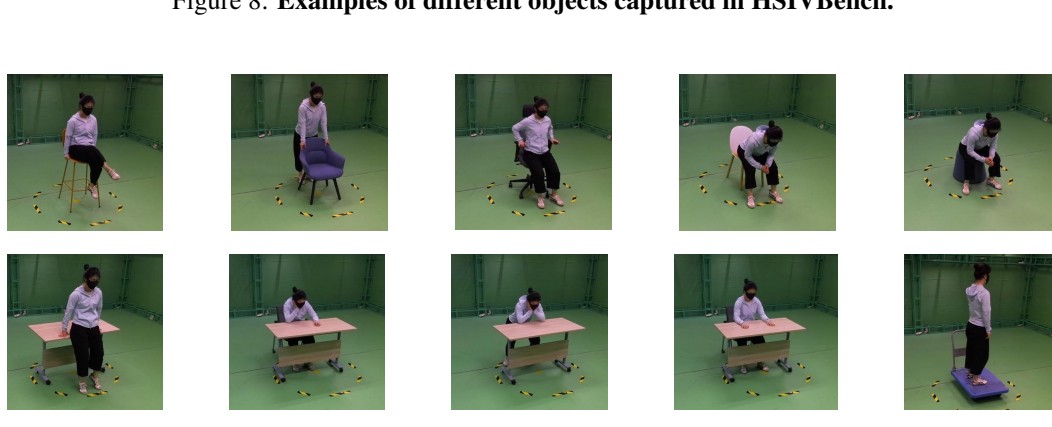

Figure 9: **Examples of female data in HSIVBench.**

# B  FURTHER ANALYSIS

**Analysis of the Number of Input Views.**    In Tab. 3, we analyze the effect of the number of input views. The results indicate that increasing the number of views leads to slight improvements in human motion quality. Interestingly, however, we notice that the number of views has little impact on penetration handling or the overall stability of the simulation.

Table 3: **Quantitative comparison regarding simulation stability of human-scene-interaction and the quality of human motions.**

| Method | Stability-HSI (%) ↑ | | | Scene Penetration - 3D (%) ↓ | Human Motion Quality ↓ | |
|--------|------|--------|------|------------------------------|-----------|-----------|
|        | Easy | Medium | Hard |                              | W-MPJPE ↓ | PA-MPJPE ↓ |
| 16-view | **55.16** | 29.51 | 13.59 | 21.81 | **4.01** | **1.99** |
| 10-view | 52.93 | **32.17** | 13.03 | **21.00** | 4.06 | 2.05 |
| 4-view | 53.68 | 30.56 | **13.92** | 22.90 | 4.09 | 2.17 |

**Issues with HSfM Reconstruction.**    Fig. 10 illustrates the common issues of HSfM, inherited from DUSt3R. We could observe that it often fails to reconstruct object structures accurately, producing results with numerous holes that hinder the modeling of human–scene interactions.

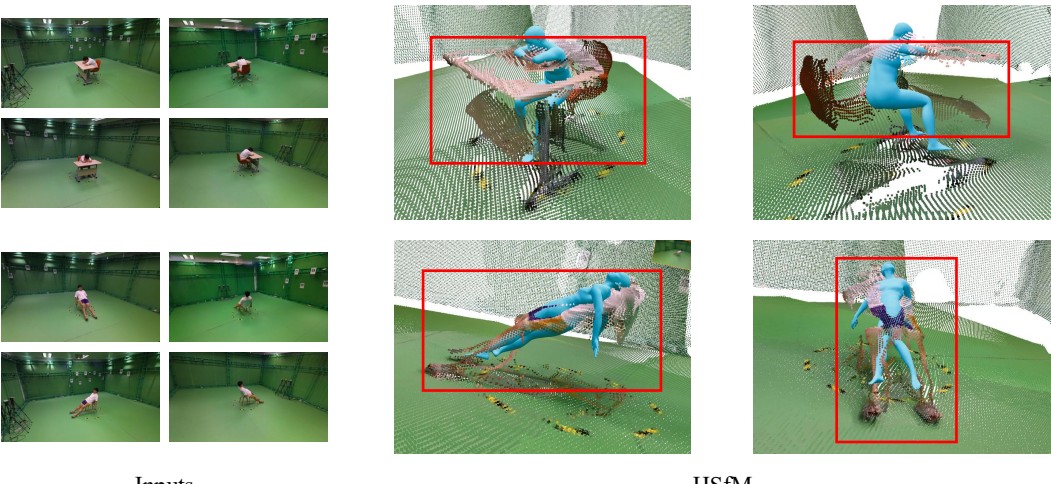

Inputs                                                          HSfM

Figure 10: **Issues with HSfM reconstruction.**

## C  LIMITATIONS AND SOCIETAL IMPACTS

**Limitations**    While HSImul3R represents the first attempt at simulation-ready reconstruction of human–scene interactions, we do acknowledge that HSImul3R has certain limitations: **(1)** the successful ratio is not very high, particularly in scenarios involving complex interactions or multiple objects (more than three); **(2)** In many failure cases, the humanoid and objects tend to end up standing independently rather than engaging in meaningful interactions (see Fig. 11); **(3)** Our fine-tuned image-to-3D model inevitably inherits biases from both the MIDI original training dataset and our collected HSIVBench, which may constrain its generalizability to out-of-domain cases.

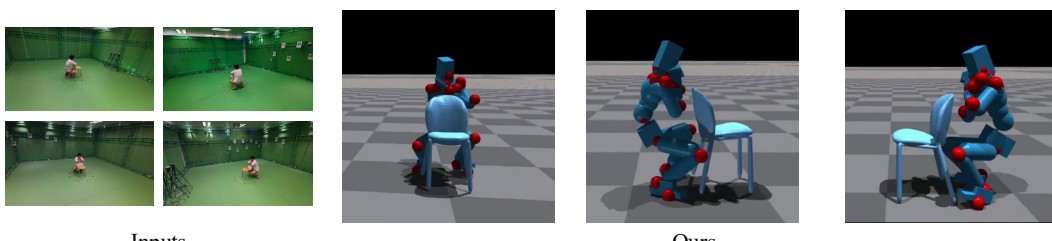

Inputs                                                                 Ours

Figure 11: **Example of failure cases.**

**Societal Impacts**    Advances in simulation-ready reconstruction, along with future research in embodied AI, hold great promise for supporting human daily life. However, they also raise concerns regarding safety and reliability. Robots or AI systems that rely on imperfect reconstructions may behave unpredictably in real environments, potentially leading to unsafe interactions or even physical harm to users. These risks highlight the importance of ensuring robustness, stability, and rigorous safety evaluation before deployment in human-centered applications.

