# OpenReview forum: "HSImul3R: Reconstructing Simulation-Ready Human-Scene-Interaction from Sparse Views"
_ICLR.cc/2026/Conference — ICLR 2026 Conference Withdrawn Submission_

### Official Review · Reviewer_q9gg · 2025-10-30

**Soundness:** 3
**Presentation:** 2
**Contribution:** 3
**Rating:** 4
**Confidence:** 4

**Summary:**

HISmul3R is proposed to reconstruct simulation-read 3D human-scene interactions from sparse-view images. Built upon HSfM, explicit 3D structural priors from image-to-3D models and contact constraints between humans and scenes are injected. Then, an RL-based refining procedure is introduced to optimize the human pose. Finally, DSRO is designed to optimize the object geometry.

**Strengths:**

- The proposed pipeline is reasonable and effective, especially the DSRO part. Comprehensive experiments are conducted to validate the effectiveness of different modules.

- The collected HSIBench could be a valuable data contribution.

- The performance of the proposed pipeline is impressive.

**Weaknesses:**

- Some technical details are missing, like the difficulty classification criteria and the training of scene-targeted RL.

- In Sec. 3.4, the three criteria of DSRO are still ambiguous, especially (2) and (3).

- According to Figure 2, (Sec. 3.2 + Sec. 3.4) appear to be conducted simultaneously with Sec. 3.3. Will the object reconstruction be influenced by the inaccurate and unstable human pose before simulation?

- The pipeline description is ambiguous or lacking support for certain scenarios. As I understand, the object is not put into the simulator in the Section. 3.3. If so, how would the lying-in-couch case in Figure 7 be processed? If I misunderstood and the object was put into the simulator, does it mean that Sec. 3.3 and Sec. 3.4 were conducted jointly? Also, the physics material parameters and densities of the object could be critical.

- Quantitative analyses of object poses are missing.

- The evaluation is unclear. In Table 1, since the human poses are different, why would the penetration not change for V2-V4?

- It might be better to replace V1-4 in Table 1 with abbreviations of the corresponding variances for clarity. Adding simple explanations in the table caption would also be helpful.

**Questions:**

- How long does it take to reconstruct a case?

- Is Eq. 5 further formulated as an extra reward of PHC? If so, the reward formulation should also be clarified.

- How many steps does the scene-targeted RL take? Will the RL stop after a certain number of steps or after some criteria are met? Is it trained from scratch or from an off-the-shelf PHC checkpoint?

- In the video, some shakes could be observed in the human. Will this also happen for scene-targeted RL? If it happened, what would be treated as the output human pose at this stage?

---

### Official Review · Reviewer_oiro · 2025-10-30

**Soundness:** 2
**Presentation:** 1
**Contribution:** 1
**Rating:** 2
**Confidence:** 5

**Summary:**

The paper presents HSImul3R, a framework for simulation-ready 3D reconstruction of human–scene interactions (HSI) from sparse-view RGB images. It builds upon HSfM (Human-Scene Structure from Motion) and introduces three new components to improve physical plausibility.
First, a Contact-Aware Interaction Modeling module integrates the image-to-3D generative model MIDI and enforces geometric consistency between human and scene via contact and non-contact losses that reduce penetration and encourage realistic coupling.
Second, the framework adapts the Perpetual Humanoid Control (PHC) controller to refine reconstructed static poses in simulation, introducing an additional scene-proximity loss that aligns human contact points with nearby object surfaces for stable contact under gravity.
Third, a Direct Simulation Reward Optimization (DSRO) stage fine-tunes the generative model using feedback from physics simulation: scenes that remain stable under contact are rewarded, while unstable ones are penalized, thus biasing generation toward physically valid geometries.
The paper also introduces HSIBench, a small benchmark dataset of 300 multi-view recordings (3 subjects, 19 household objects) for evaluating physically consistent human–scene reconstruction. Experiments show that HSImul3R achieves substantially higher physical stability and lower penetration compared to HSfM and other baselines.

**Strengths:**

1. The paper tackles a promising direction of making 3D human–scene reconstruction simulation-ready, bridging perception and physics.

2. The idea of incorporating simulation feedback into the reconstruction loop is potentially impactful.

3. The introduction of the HSIBench dataset provides a useful benchmark for evaluating physically consistent human–scene interactions.

**Weaknesses:**

1. DSRO is heuristic and weakly grounded. The proposed simulation reward optimization simply flips the diffusion loss sign based on a binary “stable/unstable” label from simulation, producing noisy, non-directional gradients that push samples away from unstable regions without indicating how to achieve stability. There is no theoretical justification or ablation showing that this mechanism learns a meaningful physical prior.

2. Static-interaction assumption and questionable use of RL. All experiments focus on static resting poses (sitting, leaning, standing), where full physics simulation is unnecessary and simple geometric consistency checks could suffice. Forcing contact in every case often results in unnatural over-contacted configurations. Moreover, using reinforcement learning to refine a single static frame in simulation is conceptually misplaced. The supplementary videos also show noticeable jittering and unnatural ostures, suggesting limited realism and stability.

3. Pipeline over-integration with limited novelty. The overall framework stitches together existing components (DUSt3R, HSfM, MIDI, PHC) with additional losses, but lacks a clear algorithmic innovation.

4. Missing discussion of relevant prior work. The paper overlooks several closely related studies in human–scene reconstruction and contact modeling, such as Resolving 3D Human Pose Ambiguities with 3D Scene Constraints (ICCV 2019), Learning Motion Priors for 4D Human Body Capture in 3D Scenes (ICCV 2021), Generating 3D People in Scenes without People (CVPR 2020), and Synthesizing Long-Term 3D Human Motion and Interaction in 3D Scenes (CVPR 2021). Similarly, the use of RL for motion refinement has been explored in SFV: Reinforcement Learning of Physical Skills from Videos (SIGGRAPH ASIA 2019) and VideoMimic (CoRL 2025), which are relevant but not discussed.

5. Insufficient quantitative analysis. The paper reports limited numerical evaluation, lacking comparisons with recent human–scene estimation methods and missing standard comparisons such as human pose error and scene reconstruction error. The evaluation focuses narrowly on stability percentages, leaving uncertainty about the method’s actual reconstruction accuracy and generalization capability.

**Questions:**

Why not to extend to 4D reconstruction?

---

### Official Review · Reviewer_YWhf · 2025-10-31

**Soundness:** 3
**Presentation:** 2
**Contribution:** 3
**Rating:** 4
**Confidence:** 4

**Summary:**

This paper proposes a method for obtaining simulation-ready 3D human-object reconstructions. The idea is interesting and the paper is overall complete, with clear figures and tables, and demonstrates the method’s effectiveness on one dataset. However, several technical details are missing, and the scope of evaluation are limited.

**Strengths:**

1. The paper presents a good idea to produce simulation-ready reconstructions that bridge human-scene estimation and physical simulation.

2. The paper is complete and reasonably well organized; figures and tables are clear and easy to follow.

3. Experiments on one dataset demonstrate the method’s potential effectiveness.

**Weaknesses:**

1. The paper mentions “While PHC was designed for dynamic motions, we adapt it to static poses by replicating each pose across the temporal dimension to fit the network.” This adaptation seems inefficient and might not be the best way to handle static inputs.

2. Figure 2 appears to contain labeling errors: loss “non-contrast” and “contrast” seem to correspond to Eq. (3) and Eq. (4), respectively, which should be "contact"?

3. The paper lacks clarity and important implementation details.

   - For example, in Eq. (2), when the scene contains $O$ objects, it is unclear how the contact-related losses are computed. Presumably, one needs to first determine whether each object has contact with the human, but this process is not described.

   - Related work on similar datasets is insufficient, and experiments are conducted on only one dataset. It would be beneficial to test on additional datasets such as CHAIRS (ICCV’23) or CORE4D (CVPR’25).

   - Some of the claimed contributions appear to be incremental, eg, adding additional loss terms to existing techniques. While this leads to some improvement, the conceptual contribution seems limited.

**Questions:**

In the demo video, is the shaking humanoid supposed to show the optimization process within the simulation environment? Without narration or explanation, it is a bit difficult to understand.

---

### Official Review · Reviewer_G5vk · 2025-10-31

**Soundness:** 3
**Presentation:** 3
**Contribution:** 3
**Rating:** 4
**Confidence:** 3

**Summary:**

This paper recovers simulation-ready 3D reconstruction of human-scene interactions from 4 uncalibrated sparse-view images. First, the 3D scene and human pose are reconstructed via HSfM. The objects in the scene are segmented and replaced with generated image-to-3D meshes produced by MIDI. A contact objective is used to enforce non-interpenetration while encouraging that the human body part nearest to the object is in contact. Finally, the human pose is retargeted to IsaacGym simulation via PHC. To improve the stability of the reconstructed 3D meshes, DSRO is used to fine-tune MIDI to produce more stable 3D meshes (which remain upright and stable while experiencing interaction throughout the simulation).

**Strengths:**

- The problem of stable human-scene reconstruction from 4 sparse-view images is an important problem.
- The proposed innovations (3d mesh reconstruction, contact optimization, and simulation reward fine-tuning of MIDI) are well-motivated and effectively help improve the stability of the human-scene reconstruction.

**Weaknesses:**

- Although the pipeline works, it does not have a very high success rate (especially on medium and hard interaction scenarios).
- The presentation of the qualitative results could be improved (e.g. explaining what the demo video is showing, and removing the "Visual Studio Code is not responding" screens from the demo video)
- The collected HSIBench dataset seems to be missing important details in the paper (e.g. how are the ground-truth geometry and human motions obtained for the evaluation in Tab. 1 and Tab. 2? What modalities are collected - e.g. RGB or RGBD? )
- Although the paper shows results on the self-captured HSIBench dataset, qualitative video comparisons of the proposed method compared to baselines (such as HSfM) would help illustrate the performance of the method.

**Questions:**

- What is the supplementary demo video showing? Why is the humanoid moving in the supplementary demo video when the reconstruction is of a static scene? Is it showing the retargeting or contact optimization over time?
- How is the Stability-HSI metric defined? Is it the same as the DSRO optimization objective? If so, it is expected that fine-tuning on DSRO directly would improve Stability-HSI metric.

---

### Note · Authors · 2025-11-13

I have read and agree with the venue's withdrawal policy on behalf of myself and my co-authors.